# First Equine Herpes Myeloencephalopathy (EHM) Outbreak in Chile

**DOI:** 10.3390/ani15162344

**Published:** 2025-08-11

**Authors:** María Carolina Durán, Macarena Suazo, Antonia Maturana, María Paz Vargas, Alfonso García, Carolyn Ahumada, Alexza Pezoa, Lutz S. Goehring, Felipe Lara

**Affiliations:** 1Unidad Medicina y Cirugía Equina, Escuela Medicina Veterinaria, Facultad de Ciencias de la Vida, Universidad Andrés Bello, Camino Lo Pinto Km 2,5 Colina, Santiago 9340000, Chile; especialistaequinos@gmail.com (M.C.D.);; 2Private Practitioner, Santiago 7591110, Chile; 3Laboratorios de Sanidad Animal, Departamento Red SAG de Laboratorios, Servicio Agrícola Ganadero, Santiago 7500007, Chile; 4Department of Veterinary Science, Gluck Equine Research Center, University of Kentucky, 1400 Nicholasville Rd., Lexington, KY 40546, USA; l.goehring@uky.edu

**Keywords:** EHM, EHV-1, herpes virus, myeloencephalopathy, outbreak, polo horses

## Abstract

Equine herpesvirus type 1 infection has only been associated with respiratory disease and abortion in Chile. Herein we describe the first outbreak with Equine Herpes Myeloencephalopathy in Chile. The outbreak affected 567 horses in a polo facility. A total of 13 horses showed signs of illness, including fever and neurological signs. The death rate was very low, and most horses recovered. Quick quarantine and strict biosecurity helped control the viral spread. Ongoing research will help us better understand and prevent future outbreaks.

## 1. Introduction

Equine herpesvirus type 1 (Equid alphaherpesvirus 1; EHV-1) is a highly prevalent virus in the equine population worldwide. Economic consequences due to EHV-1 infection are devastating for the horse industry, especially if associated with large-scale outbreaks of abortions, perinatal foal mortality, and/or myeloencephalopathy (EHM).

EHV-1 and EHM outbreaks are frequently reported in the northern hemisphere (North America and Europe) [1] and somewhat rare in the southern hemisphere. In South America, EHM outbreaks have been reported in Brazil [2] and Argentina [3,4]. Nevertheless, to date, EHM cases have not been reported in Chile. Evidence that could explain this phenomenon is lacking; it could be related to poor disease surveillance and reports as well as viral, environmental, and individual factors.

The present report describes clinically and epidemiologically the first EHV-1 outbreak with EHM in a Polo Club/Training center in Santiago, Chile.

## 2. Materials and Methods

### 2.1. Horses, Premises, Outbreak Management

#### 2.1.1. Horses, Premises, Initial Events

The outbreak occurred in the summer of 2025 (end of February) at a Polo operation located in Santiago, Chile. The “Club de Polo Golf Lo Recabarren” (CDP) has a capacity of approximately 800 hoses, with 10 barns (80 stall) and 1 roofed-pens barn (Figure 1). At the end of February, coinciding with the beginning of the Polo season, each barn housed approximately 55 horses from various regions of Chile, with most brought to the CDP a few days prior to the first neurological case. All horses were vaccinated against influenza (EQUILIS^®^ PREQUENZA-Te, Merck & Co., Inc., Rahway, NJ, USA imported by Intervet Chile Ltda., Santiago, Chile.; Cabolan^®^, Veterquimica S.A., Santiago, Chile). Nevertheless, the majority of the horses were not vaccinated against herpes virus. Only 10 horses had a history of previous herpes vaccination (Fluvac Innovator EHV 4/1^®^, Zoetis, Santiago, Chile).

The Club de Polo Golf Lo Recabarren (CDP) is located in Santiago, Chile. It is a privately owned club that houses only polo horses. The nearest equestrian facilities are approximately 2 km away. Barns are labeled with letters—A, B, C, D, E, F, G, H, I, J, and RP (roofed pens)—with V: Vet Care Office/Pharmacy; P: Parking; K1 and K2: Polo Fields; and L: Training area. Red cross: Entrance. Red Rectangle: isolation area, Barn D.

On 22 February (see Figure 2, outbreak timeline), a polo match was held 170 km south from the CDP (near the city of Chimbarongo in the O’Higgins region). Horses from four different owners were brought together at the match. Two owners brought their horses to the CDP after the match, while the horses of the other two owners remained in the region.

Three days later (25 February), a mare housed at barn D of the CDP developed acute hind limb ataxia (index case) and was treated with NSAIDs, corticosteroids, vitamins, and IV-fluids. On the morning of 26 February, the mare was found in lateral recumbency and euthanized with no necropsy performed. The mare was purchased in Los Ángeles (Bío-Bío region, southern Chile) in January and transported from Chimbarongo to CDP on 23 February after the above-mentioned polo match.

The same day the mare was euthanized (26 February), a polo match was held at CDP with local horses and horses brought to CDP for the match. In the afternoon a second mare (case 1), part of a group of horses that played at the CDP polo match, showed similar neurological signs (acute ataxia and recumbency). Infectious causes of neurological disease were suspected (EHV-1, viral encephalomyelitis, WNV, etc.) and the national animal disease agency (Servicio Agrícola y Ganadero, SAG) was notified as most of the differential diagnoses require notification in Chile. SAG-personnel tended to the CDP operation and collected samples; contact horses were isolated, and a quarantine of the entire CDP was initiated (Figure 2). Afterwards, the mare (case 1) was also euthanized due to poor treatment response and prognosis.

From 28 February to 1 March, three new horses, also owned by the index case owner, developed ataxia. All horses with neurological signs were treated with IV-fluids, anti-inflammatory drugs, and multi-vitamin complexes (vit E, B, C, and selenium).

On 2 March, SAG reported the presence of Equine Herpes Virus 1 in the samples collected from case 1 and contact horses, confirming the suspected Equine Herpes Myeloencephalopathy (EHM) outbreak. Biosecurity measures were reinforced, a 28-day quarantine was determined, and horse movements from and to the premises were stopped.

A few days later (9 March), a horse in Chimbarongo that participated in the polo match on 22 February was reported to show ataxia. The horse was isolated, tested, and treated. On 10 March, another horse there also showed neurological signs, thus it was also isolated, tested, and treated. Both horses were also EHV-1 PCR positive on nasopharyngeal swabs.

The following days, all horses were closely monitored—rectal temperature was measured twice daily to detect and isolate clinical cases (fever and/or EHM)—and treated promptly.

#### 2.1.2. Horse Movements and Biosecurity

Prior to quarantine, there was neither thorough examination of the horses on arrival to the CDP, nor the recommendation to measure the rectal temperature of all housed horses and participants once or twice daily. Management, training, and competition meant that horses routinely crossed each other, and cats and dogs were running freely outside and inside the barns.

Once EHM was suspected, strict biosecurity measures were applied. The index case barn (barn D) was isolated. When the EHV-1 diagnosis was confirmed by the local animal health officers (SAG), a 28-day mandatory quarantine was established. Unfortunately, although horse movements from and to the premises were stopped, horses were moved within the premises depending on their history of possible contact with PCR-positive horses (Figure 3). The horses of the owner of the index case and case 1 were kept in isolation (barn D), while the other horses from barn D were moved to barn C. Horses in barn C were moved to barn J, except for the horses of one owner, whose horses played at the match in Chimbarongo, that were moved to barn F. One of these horses developed ataxia and was moved to isolation (barn D); the other horses of this owner were moved to RP (Figure 3).

Physical exams of all isolated horses were performed daily (heart rate, respiratory rate, rectal temperature, presence of respiratory or neurological signs, evaluation of mucous membranes, and presence of gastrointestinal motility were recorded). Sick horses were more frequently assessed depending on their health status. All data were recorded in clinical records. If fever was detected, standardized treatment was initiated with NSAIDs and vitamin complexes, and rectal temperature was measured every 1–2 h until normalization. Ataxia was assessed and classified according to the scale described by Olsen et al. [5], which grades the severity of neurological signs from 0 to 5.

General biosecurity measures included the controlled movement of personnel and horses within the operation. Contact among horses from different barns was prohibited, the daily exercise routine (30 min walks twice daily) was strictly scheduled and regulated to prevent commingling/crowding, shared water troughs were eliminated, stalls were cleaned daily and disinfected with quaternary ammonium chloride (10%) and hypochlorite dilutions (0.1%), and rectal temperature was taken twice daily. If hyperthermia was detected, the horse was immediately treated and isolated in Barn D (Figure 1 and Figure 3) to facilitate medical treatment, and the official veterinarian from SAG was notified for further testing.

A veterinary team was established within the isolation area to implement and monitor clinical treatments. Strict biosecurity measures were applied both during intra- and inter-patient management. In barn D, the physical separation of each horse was also established, horses were grouped according to their symptoms (fever/neurological signs), and raschel mesh barriers/fences were installed between stalls to reduce the risk of airborne transmission.

Treatment was divided into two protocols based on clinical signs. Horses exhibiting only hyperthermia were treated primarily with NSAIDs (flunixin meglumine) and vitamin complexes. In cases where progression to neurological signs was evident, the administration of corticosteroids (dexamethasone) and DMSO diluted in saline solution was included. In some cases, the antiviral acyclovir was administered. A good quality diet, free access to water, and nursing care were provided. If horses were not able to stand, a hoist with padded slings was used to lift and provide standing support. Treatment duration depended on the patient’s clinical condition but also the owner’s available budget. Treatments were adjusted according to each patient’s clinical progress, with modifications primarily aimed at gradually reducing dosing or withdrawing the drug based on the observed therapeutic response.

### 2.2. Sample Collection, Diagnostic Testing

Official veterinarians from the animal health office (SAG) collected all samples for laboratory analysis. Initial post-mortem samples were collected on February 27, from case 1 (tracheal secretions, brain tissue, blood samples). Simultaneously, blood samples were collected from the other 12 horses housed in the same barn as case 1 and owned by the same owner. On 3 March, nasopharyngeal swabs were collected from the same 12 horses. The procedure was carried out using sterile uterine swabs, which were introduced through the ventral nasal meatus until reaching the nasopharynx. Once in position, the swab tip was exposed and briefly maintained in contact with the mucosa to collect the sample (retropharyngeal sampling). The swab was then retracted, placed in a sterile tube, and immediately suspended in viral transport medium. Samples from the index case could not be taken as the mare had been euthanized and buried. Trauma was suspected, and it was thought to be an isolated case; once more neurological cases started to appear, infectious diseases were suspected, and investigations started.

All samples were tested for infectious neurological diseases (EHV-1 PCR/SN, Eastern Equine Encephalomyelitis PCR/IHA, Western Equine Encephalomyelitis PCR/IHA, Venezuelan Equine Encephalomyelitis PCR/IHA, West Nile Fever PCR/ELISA, and Listeriosis culture) and for other diseases that were included by the local authorities (Equine Infectious Anemia Coggins Test, Equine Viral Arteritis PCR/ELISA, and Glanders—Burkholderia mallei CF).

### 2.3. EHV-1 Quantitative-PCR

Samples used for EHV-1 PCR analysis were the nasopharyngeal swabs, while tracheal content and cerebral tissue were collected from case 1.

Viral nucleic acids were extracted from blood, nasal secretions, and tissue samples, following manufacturer’s instructions (Mag Max Core Nucleic Acid Purification Kit, ThermoFisher Scientific, Waltham, MA, USA). All samples were subsequently assayed for the presence of the glycoprotein B (gB) gene of EHV-1 by quantitative-PCR as described previously by Diallo et al. [6]. Briefly, reactions were performed with the AgPath-ID™ One-Step RT-PCR kit (Applied Biosystems, Foster City, CA, USA) and custom primers/probe. Each 25 µL reaction included 1 × buffer, 2 mM dNTPs, 1.5 mM MgCl_2_, 0.4 µM primers, 0.1 µM probe, and 2 µL of RNA. Cycling conditions were as follows: 45 °C for 10 min, followed by 95 °C for 2 min; then, 50 cycles of 95 °C for 15 s and 64 °C for 1 min. Fluorescence was acquired at each extension. Reactions were run in single replicate; CT ≤ 40 was considered positive.

Results reported in CT (cycle thresholds).

### 2.4. EHV-1 Seroneutralization

The standard seroneutralization (SN) test was performed in serum samples collected from some cases at the start of the disease investigation. The SN test was performed as described in the Manual of Diagnostic Tests and Vaccines for Terrestrial Animals. 13th ed. 2024. Chapter 3.6.8. Equine rhinopneumonitis (infection with Varicellovirus equidalpha1) [7]. Neutralization titers were expressed as log2 of the highest dilution inhibiting cytopathology.

### 2.5. Data Analysis

Descriptive analysis of data collected from the first Chilean EHM outbreak is provided. Statistical evaluation was performed by IBM SPSS software (version 30, College Station, TX, USA). Frequency tables were used for descriptive analysis of categorical variables. The chi-square test and Cramer’s V were used to test relationships between categorical variables; *p*-values of <0.05 were considered statistically significant.

## 3. Results

### 3.1. Horses, Premises, Outbreak Management

A total of 567 high performance polo horses were kept at the CDP operation, 58.4% were mares and 41.6% geldings and stallions.

Mean age was 9.7 ± 0.132 years, ranging from 4 to 22 years of age. Mean age of horses that developed neurological signs was 9.5 ± 2.5 years ranging from 7 to 13 years.

Horses were owned by 61 different owners, each in charge of groups of 5–10 horses. Before the outbreak, training and management of each group was variable, mostly depending on owner/trainer preference. Regarding vaccination status, only a few individuals were reported to have been vaccinated against EHV-1 (three owners, approximately 10–15 horses).

Most horses arrived at the CDP facilities just before the beginning of the Polo season in Chile (end of February 2025). Majority of horses came from the Metropolitana (44.6%) and O’Higgins (30%) (Figure 4).

Traceability results showed a total of 19 direct contacts with the index case and case 1 (3.4%).

#### 3.1.1. Clinical Signs

A total of 13 horses showed clinical signs compatible with EHV-1 infection, 8 had fevers (1.4%) and 11 developed EHM (neurological signs—ataxia/recumbency with normal mentation) (1.9%) (Table 1). Only 6/11 horses with EHM developed neurological signs and high fevers (>39.5 °C). The mortality rate was 0.35% (two horses—index case and case 1) while morbidity 3.35% (19/567 horses had a positive PCR and/or clinical signs—fever/ataxia).

Fever was defined as an elevated rectal temperature above 38.5 °C at more than one time point. Of the eight horses that developed fever, three developed signs of ataxia afterwards, while another three developed fever and ataxia at the same time. The remaining two only had fevers without progression to neurological signs. Six horses showed neurological signs without developing fevers at any point of time during the clinical course.

A total of 11 horses presented some degree of ataxia: 6 cases corresponded to grade 3, 2 to grade 4, and 3 to grade 5 (Table 1). In addition, one case of abortion was recorded in a mare where gestation had not been previously diagnosed. Gestation length was estimated at approximately three months, consistent with early pregnancy loss (first trimester). No further testing was performed on this mare or fetus. In three cases, urinary incontinence associated with neurological signs was observed. One horse developed corneal ulcers, attributable to episodes of recumbency.

Blood work (complete blood count and biochemical panel) was performed in a small number of clinical cases (*n* = 13). Only mild abnormalities were detected. Four horses had anemia, three had changes in red blood cell morphology, and four had lymphopenia or leukopenia. Five horses had hypocalcemia, three had elevated AST, two had hyperbilirubinemia, two had hypophosphatemia, and one had hyperphosphatemia.

#### 3.1.2. Treatment

Horses with fever and/or neurological signs or direct-contact-horses received treatment (24 horses (4.2%)), but acyclovir was only added to the treatment in a quarter of the cases (6 horses, 5/6 showing neurological signs).

Treatment for horses that developed fevers was based on NSAIDs (flunixin meglumine 1.1 mg/kg BID IV) and vitamin supplementation (TRM vitamin E 2250 IU and selenium 500 mcg PO SID; vitamin B1 220 mg, B6 220 mg, and B12 21 mg IV SID; vitamin C 4 g IV). If the horse developed neurological signs, such as ataxia, treatment with corticosteroids (dexamethasone 0.1–0.2 mg/kg SID IV) and DMSO (50 mL 10%DMSO diluted in 3 L of LRS administered IV) was added for 1–11 days (average 5.3 days) depending on clinical response.

Acyclovir was added to the treatment once it became available. Limited access to the drug affected both the timing and duration of its use. In some cases, it was administered at the onset of neurological signs; in two cases, after clinical signs had progressed; and in only one case, at the onset of a fever (Table 2).

Out of the 13 horses that developed clinical signs (fever and/or neurological signs), only 6 were treated with acyclovir (4500 mg, three times a day, IV), 5 of which had signs of ataxia and 1 with only a fever. The duration of acyclovir treatment varied among individuals, with a maximum of 8 days in one case and a minimum of 1 day, mainly due to financial constraints and drug availability (Table 2). Additional nursing care treatments were added as needed. Thirteen direct contact horses with the index case and case 1 (and owned by the same owners) received supportive treatment that consisted of the administration of flunixin meglumine (1.1 mg/Kg EV, SID) and vitamins (TRM vitamin E 2250 IU and selenium 500 mcg PO SID; vitamin B1 220 mg, B6 220 mg, and B12 21 mg IV SID; vitamin C 4 g IV) for 5–10 days. These horses were kept in the isolation barn (D) and in RT. Positive PCR results were obtained in three of these asymptomatic isolated cases.

### 3.2. Sample Collection, Diagnostic Testing

Initial EHV-1 PCR analysis was performed on case 1 and contact or suspect horses (*n* = 21). Horses from the first two affected owners, as well as horses that had direct contact with symptomatic horses, were included in EHV-1 PCR analysis. Two contact horses owned by owner 1 and one contact horse owned by owner 2 were PCR-positive. Out of these PCR-positive contact horses, two became symptomatic (from owner 1), while the contact horse from owner 2 stayed asymptomatic.

During quarantine, PCR analysis was performed in a total of 214 horses (37.7%). It was not possible to sample all horses at the facility due to financial constraints. From the analyzed animals, 201 were PCR negative for EHV-1 (93.9%) and 13 positive (6.1%) (Table 2). Of the 13 PCR-positive horses, 7 horses showed clinical signs (53.8%), while the remaining 6 were asymptomatic. Of these asymptomatic horses, three belonged to owners 1 and 2, and the remaining three had not been in direct contact with the index case. In contrast, 6 of the 201 horses that tested negative for the EHV-1 PCR analysis, showed clinical signs compatible with EHV-1 infection (either fever and/or ataxia) after sample collection, although their PCR results were negative (2.9%).

Cycle Threshold (CT) of positive PCR results ranged from 30 to 39. Higher CT results came mostly from asymptomatic cases; a CT of 30 was obtained when brain tissue of case 1 was analyzed.

EHV-1 viral seroneutralization was performed in a small number of animals (13/567, 2.2%) at the beginning of the outbreak. Blood samples were collected from horses owned by the same owner of the index case. Of those, only one showed clinical signs compatible with EHV-1 infection. Five horses showed titers of 1:128, three of 1:64 and two > 1:256. Subsequently, out of these cases, PCR analyses from nasopharyngeal swabs were positive in 3/13 cases (with titers of 1:32, 1:32, 1:64, respectively).

#### Relationships Between Variables

Age, Sex and neurological signs: No significant relation was detected between age (mean age 9.7 ± 0.132 years and 9.5 ± 2.5 years in EHM cases; Mann–Whitney U-test *p* = 0.94) and developing neurological signs or between sex and neurological signs (1.7% males and 2.1% of females developed EHM; Pearson chi-square 0.128, *p* = 0.5).

Owner and fevers: A significant relation was detected between the owner and the detection of fever (chi square 161.5, *p* < 0.05, Cramer’s V 0.53) (given that not all horses showed fevers, and thus, the assumption of expected frequencies was violated in the chi-square test; this limitation warrants a cautious interpretation of the results).

Owner and neurological signs: A significant relation was also detected between horse owner and development of neurological signs (chi square 207.8, *p* < 0.05, Cramer’s V 0.6) (given that not all horses showed fevers, and thus, the assumption of expected frequencies was violated in the chi-square test; this limitation warrants a cautious interpretation of the results).

Origin and clinical signs (fever or neuro signs): A significant relationship was also detected between the origin of the horses and the development of clinical signs. Of all horses, 57% of horses that developed fevers came from Chimbarongo (chi square 49.145; *p* = 0.002, Cramer’s V 0.3), while 72.7% of horses with neurological signs also came from Chimbarongo (chi square 80.894; *p* = 0.000, Cramer’s V 0.38). A high number of horses came from central Chile (Figure 3), 15% from María Pinto (Metropolitana region), 11.1% Melipilla (Metropolitana region), 8.8% San Carlos (Ñuble region), 8.3% Pichidegua (O’Higgins region), 8.1% Requinoa (O’Higgins region), and 7.9% Chimbarongo (O’Higgins region).

Contact with index case/case 1 and fevers or neurological signs: A significant relationship was detected between the contact with index case/case 1 and the development of fevers (w/o contact 546 horses, w/contact 19 horses; 3 developed fevers (15.8%); chi square 34.020; *p* = 0.000, Cramer’s V 0.245) or neurological signs (6/19 developed neuro signs (31.6%), while there were only 3/19 horses (0.5%) w/o developed neurological signs; chi square 112.779; *p* = 0.000, Cramer’s V 0.447).

Fevers and neurological signs: A significant relationship was detected between the presence of fevers and the development of neurological signs (EHM) (8/567 horses developed fever, 6/8 horses with fever developed EHM; chi square 179.895; *p* < 0.001, Cramer’s V 0.563).

Positive PCR result and fevers or neurological signs: A significant relation was also detected between a positive PCR result and the development of fevers (3/13 horses with positive PCR results developed fevers—23.1%, while 4/201 horses with negative PCR results developed fevers—2%; chi square 17.160; *p* = 0.000, Cramer’s V 0.283) or neurological signs (7/13 horses with positive PCR results developed neurological signs, while 3/201 horses with negative PCR results developed neuro signs; chi square 75.131; *p* = 0.000, Cramer’s V 0.593).

### 3.3. Sequelae/Outcome

#### 3.3.1. Morbidity and Mortality Rates

From the total of 567 horses housed at the CDP, at least 19 became infected with EHV-1 (clinical cases and asymptomatic PCR positives; morbidity rate 3.35%), where only 13 developed clinical signs (of which 11 horses developed EHM (1.9%) and 2/567 horses died; mortality rate 0.35%). The remaining horses responded well to treatment and progressed favorably. At the time of discharge (22 April 2025—end of quarantine), only 3/11 EHM horses still showed neurological signs (hind limb ataxia, grade 2 and 3). The remaining horses recovered completely.

During the outbreak, the established mandatory quarantine of the CDP operation managed to contain almost completely the dissemination of EHV-1. However, two horses that were contact horses with the index case (played at the polo match on 22 February) developed ataxia on 9 March in Chimbarongo. Both horses were EHV-1 PCR positive on the collected nasopharyngeal swabs and were quarantined in their premises in Chimbarongo. They were treated by their local veterinarian following recommendations from the CDP vet team. Both horses became recumbent, and one was euthanized due to poor treatment response. The other horse recovered progressively with medical treatment. No other cases were since then reported in Chile.

#### 3.3.2. End of Quarantine and Post-Quarantine Measures

On 2 April, before ending quarantine (5 April), all horses in isolation were PCR-tested (barn D). In all other barns, 10 horses per barn were randomly PCR-tested. A total of 101 horses were tested, of which 9 tested positive and 2 were asymptomatic horses located in barns B and C. PCR-testing was repeated on 9 April; of 92 tested animals, 1 asymptomatic horse tested positive and was isolated for an additional 28 days. The remaining horses could return to their home base with biosecurity and vaccination recommendations. Horses had to remain isolated from other horses for 2 to 4 weeks, with temperature checks twice daily. Horses that developed hyperthermia had to be immediately isolated for 4 weeks. If clinically sound after 2–4 weeks of isolation, horses could be released to a pasture or resume their training/exercise schedule. Horses that recovered from EHM were not allowed to exercise for 6–12 m. Regarding vaccination, it was recommended to continue or start the EHV-1 vaccination schedule (Fluvac Innovator) 4 months after discharge, or to administer a booster when appropriate.

## 4. Discussion

This is the first report of an EHM associated with an EHV-1 outbreak in Chile. Between 1969 and 1976, the first EHV-1 outbreak in Chile was reported, mainly causing abortions [8]. Since then, isolated cases of EHV-1 abortions or respiratory disease have been reported; however, diagnosis has often been based on histopathological changes [8].

The present EHM outbreak has some interesting epidemiological aspects. It occurred in the southern hemisphere and in midsummer, although EHM outbreaks in the northern hemisphere are more commonly seen in cooler months [9,10]. Opposed to what has been previously reported [9,11], no significant association of EHM and breed, increasing age, and sex was found, but it was likely due to the type of equine operation that only housed Polo Horses in competition (Thoroughbred–Criollo-Horse crosses). Nevertheless, when EHM cases were analyzed, 7/11 EHM cases were mares (64%), but generally mares were preferred for Polo, and this was likely a confounding factor. In respect to age, mostly middle-aged horses (mean age 9.7 years, ranging from 4 to 22 years) were present at the time of the outbreak; however, EHM was detected in ages from 6 to 13 years, not only in the older horses present.

The outbreak occurred after commingling of horses in a competition, and the spread occurred in a training operation that facilitated viral spread (frequent horse movements, frequent direct and indirect contacts between animals, and high human traffic).

Although many horses were housed in the operation (*n* = 567), and practically no vaccine protection was available, only 13 horses developed clinical signs, 11 showed EHM (with or without previous fevers) and 2 only had fevers, with a morbidity of 3.35%. The proportion of horses that developed EHM was very low (1.9%) when compared with 30 to 50% reported in susceptible breeds (Thoroughbred, Standardbred, WB, and American Quarter Horses) [9,11,12,13,14]. It is well known that the spread of equine alpha herpesviruses is very difficult to prevent at large equine operations, and high morbidity rates can be seen with EHM outbreaks (>80%) [11]. Nevertheless, in the Chilean outbreak, it was possible to prevent spread of the virus out of the CDP operation to other regions. Only two EHM cases occurred outside the CDP, but both horses were in direct contact with the index case. Probably a combination of the presence of a viral variant that was not highly contagious, and the instauration of prompt and strict movement and biosecurity measures can explain successful outbreak control. With EHV-1 outbreaks, all measures should indeed be oriented to mitigate the effects of infection by rapid identification of an index case, immediate testing of contact animals, and isolation of shedding horses.

The number of infected horses was likely much higher than what could be reported because financial constraints limited testing of horses during the outbreak and before ending quarantine. Horses were tested at the beginning of the outbreak. Later, new clinical cases were considered EHV-1 infected animals and not tested. Before ending quarantine, to reduce the risk of spreading after opening the quarantine, a small number of horses per barn were randomly tested. This procedure likely left many infected asymptomatic horses undetected.

Unlike other outbreaks, in the presented case, clinical signs associated with infection were limited to fevers, ataxia, and a single abortion (no evidence of respiratory disease). Fevers did not always precede the onset of EHM. In the first two cases, rectal temperature was not regularly taken, thus fevers were likely missed. Nevertheless, during the outbreak rectal temperatures were measured twice daily in all horses and four developed EHM and never showed fever. Some horses developed fevers after starting to show EHM. Thus, in almost half of the cases high fever was not a predictor of onset of EHM. Interestingly, the abortion in a mare occurred in the first trimester of gestation and not in the last third of gestation (EHM and treatment could explain abortion, but unfortunately, abortion causes were not investigated). Absence of a fever during EHV-1 infection correlates with no or low-grade viremia [15]. However, as primary respiratory tract infections occur, horses can be asymptomatic but replicate and shed virus and seroconvert [15]. So much so that when EHM outbreaks occur, only a proportion of animals develop fever or EHM, and a subclinical infection of most of the animals can be assumed [15]. Viremia is a prerequisite for EHM (and abortion), and high fever is a risk factor for EHM [9]. However, undetectable viremia has been reported in vaccinated horses exposed to the virus [16,17]. Nevertheless, in the studied Polo population EHV1 vaccination was not commonly part of the regular vaccination schedule.

The mortality rate observed in the Chilean outbreak was 0.3%, much lower than that reported in other outbreaks such as those in the US (6.4% in 2021 and 18% in 2024 [11,18]) and Valencia (5%) [1]. This difference could be related to EHV-1 viral characteristics and prevalence or other risk factors. Nevertheless, less severe EHM outbreaks are somewhat common in the southern hemisphere when compared with the northern ones [2,4].

In general, there was a good response to treatment (based on reducing inflammation and promoting neuroprotection). The use of anticoagulants has been described, but this was not used as a baseline treatment during the outbreak due to lack of evidence and poor availability of drugs. Susceptibility of EHV to antivirals varies depending on the virus isolated and the method used by the laboratory to determine its susceptibility [19]. Acyclovir was able to stop EHV-1 replication in vitro [20] but clinical usefulness of antiviral therapy in horses was not clear [21]. In the herein described outbreak, no significant differences were observed in the resolution of neurological signs in horses treated with acyclovir. Of the three horses that still showed neurological signs at discharge, two were only given acyclovir for two days, and one did not receive antiviral therapy. Nevertheless, the effect of acyclovir could not be objectively evaluated due to the low number of cases, variable time and duration of treatment.

In the presented outbreak, virus detection was performed with EHV-1 PCR analysis, specifically EHV-1 glycoprotein B detection. Nevertheless, EHV-1 ORF30 A/G/C2254 detection was not available. From 13 positive PCR results, 7 horses presented clinical signs (53.8%), while 46.2% of PCR-positive animals were shedding the virus while being asymptomatic, highlighting the importance of isolation, detection, and strict biosecurity. Horses that tested PCR positive had very high CT values (30–39), making viral isolation and sequencing difficult. So far, the official laboratory (SAG) has not been able to sequence the virus but is working toward this goal. It is noteworthy that the index case had a history of being recently purchased and coming from a farm where a mare aborted six months earlier (no further testing for causes was performed then). It was also mentioned that at least one of the animals at that farm was imported from Argentina. Highlighting even more the need for sequencing analysis. Viral qualities, infectious dose, and innate/specific immunity in the host determine the degree of clinical signs following an infection; thus, more information on viral prevalence and type present in the Chilean horse population is required to further understand the outbreak and prepare for the impact of future outbreaks.

## 5. Conclusions

This is the first EHM outbreak reported in Chile showing low morbidity and mortality rates. Viral spread was successfully controlled with prompt and strict quarantine and biosecurity measures. More information on viral prevalence and variant type during the outbreak and in the Chilean horse population is required to further understand the outbreak and prepare for the impact of future ones.

## Figures and Tables

**Figure 1 animals-15-02344-f001:**
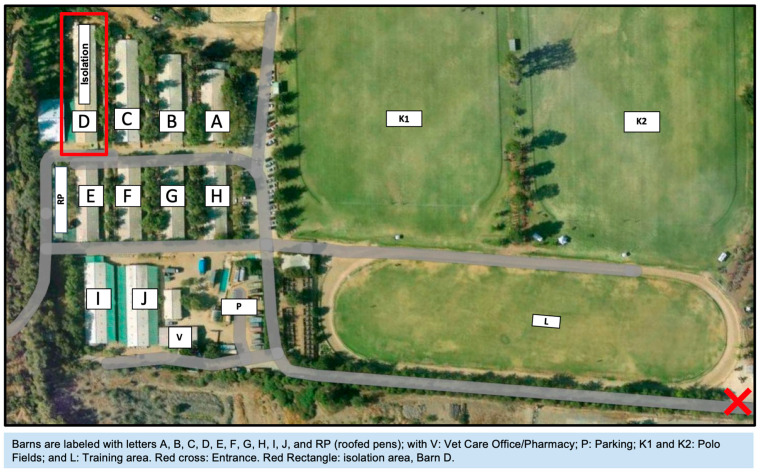
Aerial view of the Club de Polo Golf Lo Recabarren (CDP), Santiago, Chile.

**Figure 2 animals-15-02344-f002:**
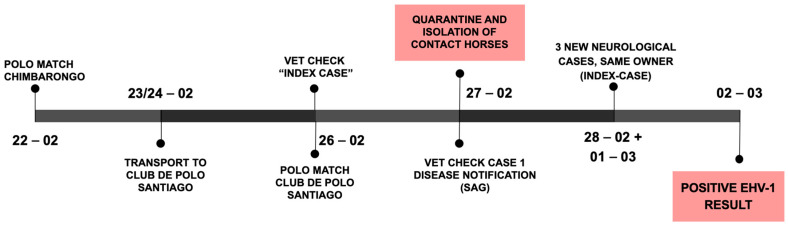
Outbreak Timeline of initial events.

**Figure 3 animals-15-02344-f003:**
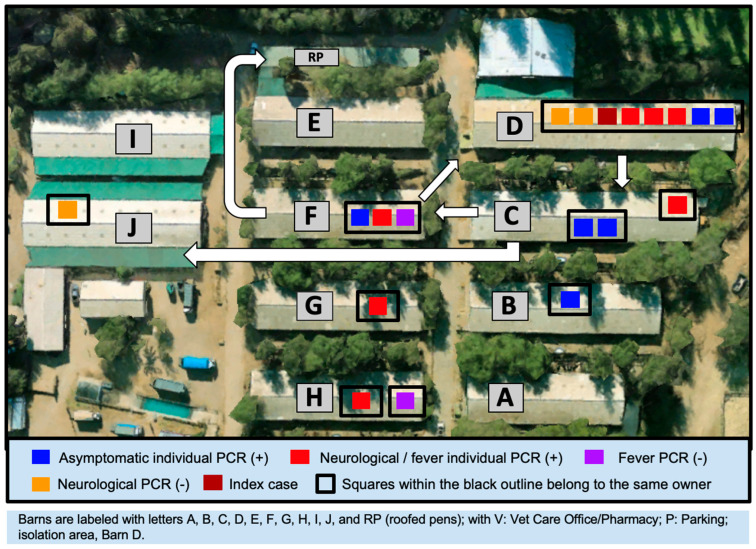
Horse distribution before the outbreak and movements within the premises depending on their clinical signs and history of possible contact with PCR-positive horses. The horses of the owner of the index case and case 1 were kept in barn D, while the other horses were moved to barn C. Horses in barn C were moved to barn J, except for the horses of one owner, whose horses played at the match in Chimbarongo, that were moved to barn F. One of these horses developed ataxia and was moved to isolation (barn D); the other horses of this owner were moved to the roofed pens (RP).

**Figure 4 animals-15-02344-f004:**
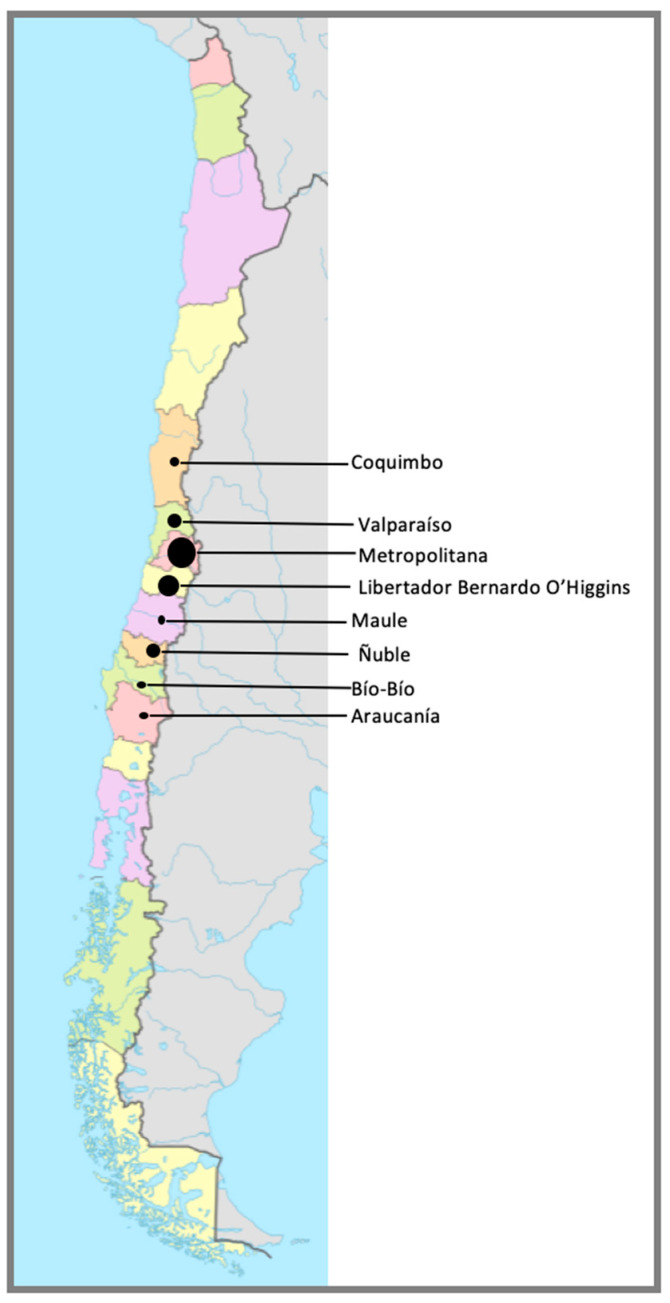
Geographic distribution of the regions of origin of horses housed at CDP at the time of the outbreak (*n* = 567). Each region is delimited by a bordering line and a different color. The size of the circle in each region (Coquimbo, Valparaíso, Metropolitana, Libertador Bernardo O´Higgins, Maule, Ñuble, Bío-Bío, Araucanía) is associated with the number of horses originating from each region—Coquimbo: 1.8%; Valparaíso: 8.6%; Metropolitana: 44.6%; Libertador Bernardo O’Higgins; 30%. Maule: 1.6%; Ñuble: 10.1%; Bío-Bío: 1.2%; and Araucanía: 2.1%.

**Table 1 animals-15-02344-t001:** Clinical signs developed by the horses during the EHM outbreak in the CDP Santiago, Chile.

Clinical Signs	Studied Population (*n* = 13)
Just Fever (>38.5 °C)	2 (15%)
Fever and ataxia	6 (46%)
Just ataxia	5 (38%)
Ataxia grade	
Ataxia 0–2	0
Ataxia 3	6
Ataxia 4	2
Ataxia 5	3
Abortion	1 (7.7%)
Corneal ulcer	1 (7.7%)
Urinary incontinence	3 (23%)

Ataxia 0: No gait deficits at the walk; Ataxia 1: No gait deficits at the walk, and deficits only identified during further testing; Ataxia 2: Deficits noted at the walk; Ataxia 3: Marked deficits noted at the walk; Ataxia 4: Severe deficits noted at the walk and may fall or nearly fall at normal gaits; Ataxia 5: Recumbent horses [5].

**Table 2 animals-15-02344-t002:** Data collected from symptomatic and/or PCR+ individuals. The following data were collected: age (years), sex (mare, gelding/stallion), origin (region), owner, barn (where they were before isolation), PCR result, contact (with at least 1 positive and/or symptomatic patient), EHM onset, fever onset, (rectal temperature > 38.5 °C), acyclovir administration, treatment duration, and survival.

Horse	Age	Sex	Origin	Owner	Barn	PCR	Contact	Fever Onset	EHM Onset	Acyclovir (E/L)	Acyclovir Tx d	Survival
1	13	M	O’H	Ow1	D	*	Yes	Unc	26/02	No	0	No
2	10	M	O’H	Ow1	D	+	Yes	Unc	27/02	No	0	No
3	10	M	O’H	Ow1	D	+	Yes	No	28/02	No	0	Yes
4	12	G/S	O’H	Ow1	D	−	Yes	No	28/02	No	0	Yes
5	12	M	O’H	Ow1	D	+	Yes	No	01/03	Yes (L)	2	Yes
6	8	G/S	O’H	Ow1	D	−	Yes	18/03	28/02	No	0	Yes
7	7	M	O’H	Ow2	F	+	Yes	06/03	03/03	Yes (L)	8	Yes
8	6	M	O’H	Ow3	C	+	Yes	05/03	07/03	Yes (L)	3	Yes
9	7	M	MET	Ow4	J	−	No	06/03	10/03	Yes (L)	5	Yes
10	12	G/S	O’H	Ow5	H	+	No	No	09/03	Yes (L)	5	Yes
11	7	G/S	O’H	Ow6	G	+	No	08/03	13/03	Yes (L)	3	Yes
12	7	M	O’H	Ow7	H	−	No	11/03	No	No	0	Yes
13	7	M	O’H	Ow2	F	−	Yes	09/03	No	Yes (E)	1	Yes
14	10	G/S	O’H	Ow2	F	+	Yes	No	No	No	0	Yes
15	9	G/S	O’H	Ow1	D	+	Yes	No	No	No	0	Yes
16	9	G/S	O’H	Ow1	D	+	Yes	No	No	No	0	Yes
17	7	G/S	MET	Ow8	B	+	No	No	No	No	0	Yes
18	12	M	MET	Ow9	C	+	No	No	No	No	0	Yes
19	10	M	MET	Ow9	C	+	No	No	No	No	0	Yes
20	10	G/S	MAU	Ow10	EXT	+	Yes	Unc	09/03	No	0	Yes
21	13	M	MAU	Ow10	EXT	*	Yes	Unc	10/03	No	0	No

Sex: M mare, G/S Gelding/Stallion; Origin: O’H O’Higgins, MET Metropolitana, MAU Maule; PCR: * no exam, + positive, − negative; EHM Equine Herpes Myeloencephalopathy; EXT Cases outside of CDP; Unc: Unconfirmed; E: Early treatment—before onset of clinical/neurological signs; L: Late treatment—while showing clinical signs; and Tx: Treatment; d: Days.

## Data Availability

The original contributions presented in this study are included in the article. Further inquiries can be directed at the corresponding author(s).

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
