# Peer review of "First Equine Herpes Myeloencephalopathy (EHM) Outbreak in Chile"

_animals, 2025, doi:10.3390/ani15162344_

Round 1
Reviewer 1 Report
Comments and Suggestions for Authors
the report seems sound and the data presented are valuable but it need to be re-organized especially the part of materials and methods and results.

needed English editing to improve the report
Author Response
REVIEWER 1 (changes in text made in red)
Comment 1:
Page 1, Line 25 Simple summary: this country change to Chile
Thanks for pointing this out, it was changed to Chile (line 25, page 1)
Comment 2:
Page 1, Line 32-34: We agree it’s a bit redundant, thus the paragraph “This is the first reported EHV-1 outbreak with EHM in Chile and South America. This study aims to describe its clinical, epidemiological, and outcome data. Clinical and diagnostic data were analyzed. EHV-1 was detected by qPCR targeting glycoprotein B. Descriptive statistics were used for categorical variables; chi-square test and Cramer’s V assessed relationships.” was deleted. And “Equine herpes myeloencephalopathy (EHM) is a neurological disease in horses caused by the equine herpesvirus-1 (EHV-1). It's a serious condition, often resulting in severe neurological deficits and can be fatal. in this report, we discuss the first outbreak of EHM in Chile (clinical signs, epidemiology, diagnosis and control measures).” was added (line 32-35, page 1).
Comment 3:
Page 1, Line 35-36: Descriptive statistics were used for categorical variables; chi-square test and Cramer’s V assessed relationships --- it can be deleted from the abstract. It was deleted from the abstract.
Comment 4:
Page 2, Line 55-58: can the authors describe EHM and genetic factors related to this phenomena “EHV-1 and EHM outbreaks are frequently reported in the northern hemisphere (North America and Europe) [1] and somewhat rare in the southern hemisphere. In South America, EHM outbreaks have been reported in Brazil [2] and Argentina [3,4]. Nevertheless, to date EHM cases have not been reported in Chile. The present report describes clinically and epidemiologically”
Thanks for the comment. There is no much evidence that explains the lack of more reports in south America, it could be related with lack of disease surveillance and reports, as well as viral, environmental and individual factors. In the case of Chile, abortion and respiratory disease were historically related to EHV1 and no real explanation for lack of neurological disease has been previously studied. The virus has yet to be sequenced and disease prevalence studied to hypothesize further possible explanations.
Thus we added: “Evidence that could explain this phenomena is lacking, it could be related with por disease surveillance and reports, as well as viral, environmental and individual factors.”(page 2, line 57-59).
Comment 5:
Page 2, Line 64-68: are the horses were vaccinated before?
Thanks for your questions. Most horses were not vaccinated against EHV1. This was added in line 70-73: “All horses were vaccinated against influenza (EQUILIS® PREQUENZA-Te, Cabolan®). Nevertheless, the majority of the horses were not vaccinated against herpes virus. Only 10 horses had history of previous herpes vaccination (Fluvac Innovator EHV 4/1)”
Comment 6:
Page 5, Line 181-183: more details about the kits and manufacturing details. All samples were subsequently assayed for the presence of the glycoprotein B (gB) gene of EHV-1 by quantitative-PCR as described previously by Diallo et al.[5].
Thanks for pointing this out. Details were added in line 214-220: “Briefly, reactions were performed with the AgPath-ID™ One-Step RT-PCR kit (Applied Biosystems) and custom primers/probe. Each 25 µl reaction included 1× buffer, 2 mM dNTPs, 1.5 mM MgClâ‚‚, 0.4 µM primers, 0.1 µM probe, and 2 µl RNA. Cycling conditions: 45 °C for 10 min, 95 °C for 2 min, then 50 cycles of 95 °C for 15 s and 64 °C for 1 min. Fluorescence was acquired at each extension. Reactions were run in single replicate; Ct ≤ 40 was considered positive. Results reported in CT (cycle thresholds).”
Comment 7:
Page 6, Line 229-230: this sentences can be removed to method section, Ataxia was assessed and classified according to the scale described by Olsen et al.[7], which grades the severity of neurological signs from 0 to 5.
We agree, thanks for pointing that out. The sentence was moved up to the methods section (page 5, line 155-156).
Comment 8:
Page 9, Line 273-274: Samples used for EHV-1 PCR analysis were nasopharyngeal swabs, while tracheal content and cerebral tissue was collected from case 1. this part cab be moved to materials and methods section, also two subtitle 1. PCR; 2. Serum neutralization test
We agree, thanks for pointing that out. The sentence was moved up to the methods section (page 6, line 203-204 and 217-218).
Comment 9:
Page 11, Line 365: This is the first report of an EHM associated with an EHV-1 outbreak in Chile. Between 1969-1976 the first EHV-1 outbreak in Chile was reported, mainly causing abortions8. Since then, isolated cases of EHV-1 abortions or respiratory disease have been reported, however, often diagnosis has been based on histopathological changes [8]. is that reference or mistake?
That’s right, indeed we forgot to add the brackets to the reference 8 in that sentence, it they were now added (page 13, line 406)
Comment 10:
Page 11: it is better to make sequence to the EHV-1 isolates to compare them with other world wide strain focusing on some genes which share in the induction of neuropathogenecity like ORF30, glycoproteins B, D
We agree. Sequencing has been attempted but has not been possible yet. Our central official lab team (SAG) is working hard to get it done, but there is very little genetic material in the collected samples. Probably because viral load in this outbreak was low (low CTs), and we have to add the fact that unfortunately the index case was not analyzed (the mare was euthanized and buried, it was thought it was an isolated case, probably due to trauma, once more cases started to appear, infectious diseases were suspected and investigations started). Orf30 PCR was requested by us, but primers were not available and purchased, unfortunately they have not arrived yet (the official lab from SAG is a governmental institution and bureaucracy can make processes sometimes a quite slow).

Reviewer 2 Report
Comments and Suggestions for Authors
This paper is the first report on EHM that occurred in Chile. EHM is an infectious disease caused by equine herpes virus type 1. EHV-1 is a common pathogen in horses, but occasionally causes serious diseases. This serious disease is abortion and EHM. EHM has been reported a lot, mainly in the northern hemisphere, but very rarely in the southern hemisphere. This paper provides valuable data on EHM. The paper is logically structured and the data is carefully described. However, there are many ambiguous parts of the description. Individual comments are described below.
L266-270: Please describe when and at what time the administration of acyclovir was administered, such as whether it was administered after neurological signs were seen or when fever was administered. Also, add the data of acyclovir administration to Table 2.
L272-274: What is preventive treatment. Describe clearly what is a preventive treatment.
L173-177: Describe when samples (blood and nasopharyngeal swabs) were collected clearly and how to collect nasopharyngeal swabs.
L276-277: Please add the date of onset to Table2. Please also add acyclovir administration. Please also add EHV-1 vaccination.
Shouldn't the fever of Individual 1, 2 be unconfirmed instead of No? In addition, fever is not just Yes, but it is better to do it on the day when fever is seen.
L281-284: the authors wrote "Of these, 3 were PCR-positive, ... asymptomatic". Indicate which individuals are three PCR positive.
L287-290: The authors wrote the number of PCR positive only. Indicate which individual s are 7 horses presented clinical signs and 6 horses with a negative PCR result showing clinical signs.
L288: The authors wrote "From the 13 positive PCR results, 7 horses presented clinical sings". Does it mean the other 6 out of 13 positive PCR results did not present clinical signs? Clarify the description.
L289-290: The authors wrote "while 6 horses with a negative PCR result ... (either fever and/or ataxia). Does it mean that 6 out of 201 negative PCR showed clinical signs?Clarify the description.
L288-290: Did the horses show clinical signs before or after sample collection?
L298: The authors wrote "PCR analysis was ...". Which DNA were used for this PCR, blood or nasopharyngeal swabs? Clarify the description.
L300-335: The authors analyzed relationships between variables. How about fever and neurological signs. Add the results of relationships between fiver and neurological signs.
L371: abortions8 is abortions [8].
L394: (>80%)11 is (>80%) [11].
L413-414: Indicate individuals in Table 2 are 4 horses which developed EHM and never showed fever.
L418-420: the authors wrote "Absence of fever ... and serocovnerted". Are these descriptions about the present data or general things? If these are not about the present data, cite reference(s). If these are about the present data, indicate which individuals in Table 2.
L429: (5%)1 is (5%) [1].
L429: The authors wrote "... related to virus type". What does "virus type"mean? EHV-1 or EHV-4 or other things? Clarify the description.
L433: "neuroprotection) The" is "neuroprotection. The".
L445-446: Why was not EHV-1 ORF30 A/G/C2254 detection available? The authors should examine the PCR and sequencing to detect ORF30 A/G/C2254.
Over.
Author Response
REVIEWER 2 (changes in text made in blue)
This paper is the first report on EHM that occurred in Chile. EHM is an infectious disease caused by equine herpes virus type 1. EHV-1 is a common pathogen in horses, but occasionally causes serious diseases. This serious disease is abortion and EHM. EHM has been reported a lot, mainly in the northern hemisphere, but very rarely in the southern hemisphere. This paper provides valuable data on EHM. The paper is logically structured and the data is carefully described. However, there are many ambiguous parts of the description. Individual comments are described below.
Comment 1:
L266-270: Please describe when and at what time the administration of acyclovir was administered, such as whether it was administered after neurological signs were seen or when fever was administered. Also, add the data of acyclovir administration to Table 2.
Thanks for pointing this out. Therefore, we have added “Acyclovir was added to the treatment once it became available. Limited access to the drug affected both the timing and duration of its use. In some cases, it was administered at the onset of neurological signs; in two cases, after clinical signs had progressed; and in only one case, at the onset of fever.” (page 9, paragraph 3, line 291-294)
Acyclovir data was added to Table 2, it also now contains clinical signs onset dates.
Comment 2:
L272-274: What is preventive treatment. Describe clearly what is a preventive treatment.
Thanks for pointing this out. A clear description of what was meant by preventive treatment was added “Thirteen direct contact horses with index case and case 1 (and owned by the same owners) received preventive treatment that consisted in the administration of flunixin meglumine (1.1mg/Kg EV, SID) and vitamins (TRM vitamin E 2250 IU and selenium 500 mcg PO SID; vitamin B1 220mg, B6 220mg, B12 21mg IV SID; vitamin C 4g IV) for 5-10 days“ (page 10, last paragraph, line 300-304).
Comment 3:
L173-177: Describe when samples (blood and nasopharyngeal swabs) were collected clearly and how to collect nasopharyngeal swabs.
Thanks for your request, this information was added (page 6, paragraph 2, line 183-195)
Official veterinarians from the animal health office (SAG) collected all samples for laboratory analysis. Initial post-mortem samples were collected on February 27, from case 1 (tracheal secretions, brain tissue, blood samples). Simultaneously, blood samples were collected from the other 12 horses housed in the same barn as case 1 and owned by the same owner. On March 3, nasopharyngeal swabs were collected from the same 12 horses. The procedure was carried out using sterile uterine swabs, which were introduced through the ventral nasal meatus until reaching the nasopharynx. Once in position, the swab tip was exposed and briefly maintained in contact with the mucosa to collect the sample (retropharyngeal sampling). The swab was then retracted, placed in a sterile tube, and immediately suspended in viral transport medium. Samples from the index case could not be taken, the mare was euthanized and buried. Trauma was suspected and it was thought to be an isolated case, once more neurological cases started to appear, infectious diseases were suspected and investigations started.
Comment 4:
L276-277: Please add the date of onset to Table2. Please also add acyclovir administration. Please also add EHV-1 vaccination.
Shouldn't the fever of Individual 1, 2 be unconfirmed instead of No? In addition, fever is not just Yes, but it is better to do it on the day when fever is seen.
Thanks for your suggestions. Date of onset of fever and neurological signs was added to table 2. Also, if acyclovir was added to the treatment, and if it was added before neurological signs onset (early treatment E) or after EHM onset (late treatment L), as well as treatment duration. The fever of individuals 1 and 2 were changed to unconfirmed as you correctly pointed out, in both horses rectal temperatures were not collected before onset of neurological signs.
|
Data collected from symptomatic and/or PCR+ individuals. The following data were collected: age (years), sex (mare, gelding/stallion), origin (region), owner, box (where they were before isolation), PCR (≥1 sample), contact (with at least 1 positive and/or symptomatic patient), EHM onset, fever onset, (rectal temperature >38.5ºC),aciclovir administration, survival |
||||||||||||
|
Horse |
Age |
Sex |
Origin |
Owner |
Barn |
PCR |
Contact |
Fever onset |
EHM onset |
Aciclovir (E/L) |
Aciclovir Tx d |
Survival |
|
1 |
13 |
M |
O’H |
Ow1 |
D |
* |
Yes |
Unc |
26/02 |
No |
0 |
No |
|
2 |
10 |
M |
O’H |
Ow1 |
D |
+ |
Yes |
Unc |
27/02 |
No |
0 |
No |
|
3 |
10 |
M |
O’H |
Ow1 |
D |
+ |
Yes |
No |
28/02 |
No |
0 |
Yes |
|
4 |
12 |
G/S |
O’H |
Ow1 |
D |
- |
Yes |
No |
28/02 |
No |
0 |
Yes |
|
5 |
12 |
M |
O’H |
Ow1 |
D |
+ |
Yes |
No |
01/03 |
Yes (L) |
2 |
Yes |
|
6 |
8 |
G/S |
O’H |
Ow1 |
D |
- |
Yes |
18/03 |
28/02 |
No |
0 |
Yes |
|
7 |
7 |
M |
O’H |
Ow2 |
F |
+ |
Yes |
06/03 |
03/03 |
Yes (L) |
8 |
Yes |
|
8 |
6 |
M |
O’H |
Ow3 |
C |
+ |
Yes |
05/03 |
07/03 |
Yes (L) |
3 |
Yes |
|
9 |
7 |
M |
MET |
Ow4 |
J |
- |
No |
06/03 |
10/03 |
Yes (L) |
5 |
Yes |
|
10 |
12 |
G/S |
O’H |
Ow5 |
H |
+ |
No |
No |
09/03 |
Yes (L) |
5 |
Yes |
|
11 |
7 |
G/S |
O’H |
Ow6 |
G |
+ |
No |
08/03 |
13/03 |
Yes (L) |
3 |
Yes |
|
12 |
7 |
M |
O’H |
Ow7 |
H |
- |
No |
11/03 |
no |
No |
0 |
Yes |
|
13 |
7 |
M |
O’H |
Ow2 |
F |
- |
Yes |
09/03 |
no |
Yes (E) |
1 |
Yes |
|
14 |
10 |
G/S |
O’H |
Ow2 |
F |
+ |
Yes |
No |
no |
No |
0 |
Yes |
|
15 |
9 |
G/S |
O’H |
Ow1 |
D |
+ |
Yes |
No |
no |
No |
0 |
Yes |
|
16 |
9 |
G/S |
O’H |
Ow1 |
D |
+ |
Yes |
No |
no |
No |
0 |
Yes |
|
17 |
7 |
G/S |
MET |
Ow8 |
B |
+ |
No |
No |
no |
No |
0 |
Yes |
|
18 |
12 |
M |
MET |
Ow9 |
C |
+ |
No |
No |
no |
No |
0 |
Yes |
|
19 |
10 |
M |
MET |
Ow9 |
C |
+ |
No |
No |
no |
No |
0 |
Yes |
|
20 |
10 |
G/S |
MAU |
Ow10 |
EXT |
+ |
Yes |
Unc |
no |
No |
0 |
Yes |
|
21 |
13 |
M |
MAU |
Ow10 |
EXT |
* |
Yes |
Unc |
no |
No |
0 |
No |
|
Sex: M mare, G/S Gelding/Stallion; Origin: O’H O’Higgins, MET Metropolitana, MAU Maule; PCR: * no exam, + positive, - negative; EHM Equine Herpes Myeloencephalopathy; EXT Cases outside of CDP. Unc Unconfirmed. E: Early treatment - before onset of clinical/neurological signs; L: Late treatment - while showing clinical signs. Tx: Treatment; d: Days |
||||||||||||
Comment 5:
L281-284: the authors wrote "Of these, 3 were PCR-positive, ... asymptomatic". Indicate which individuals are three PCR positive.
Thanks for your comment, the paragraph was changed to “Initial EHV-1 PCR analysis was performed on case 1 and contact or suspect horses (n=21). Horses from the first two affected owners, as well as horses that had direct contact with symptomatic horses were included in EHV-1 PCR analysis. Two contact horses owned by owner 1 and one contact horse owned by owner 2 were PCR-positive. Out of these PCR-positive contact horses, 2 became symptomatic (from owner 1), while the contact horse from owner 2 stayed asymptomatic.” (page 11, paragraph 1, line 308-313)
Comment 6:
L287-290: The authors wrote the number of PCR positive only. Indicate which individual s are 7 horses presented clinical signs and 6 horses with a negative PCR result showing clinical signs.
Thanks for pointing this out. We agree the paragraph leads to confusion. Therefore, it was reformulated: “During quarantine, PCR analysis was performed in a total of 214 horses (37.7%). It was not possible to sample all horses at the facility due to financial constraints. From the analyzed animals, 201 were PCR negative for EHV-1 (93.9%) and 13 positive (6.1%). (Table 2). Of the 13 PCR-positive horses, 7 horses showed clinical signs (53.8%), while the remaining 6 were asymptomatic. Of these asymptomatic horses, 3 belonged to owners 1 and 2, and the remaining 3 had not been in direct contact with the index case. In contrast, 6 of the 201 horses that tested negative for the EHV-1 PCR analysis, showed clinical signs compatible with EHV-1 infection (either fever and/or ataxia) after sample collection, although their PCR results were negative (2.9%)”(page 11, paragraph 2, line 314-322)
Comment 7:
L288: The authors wrote "From the 13 positive PCR results, 7 horses presented clinical sings". Does it mean the other 6 out of 13 positive PCR results did not present clinical signs? Clarify the description.
Thanks for your comment, we hope it was clarified with the previous response (page 11, paragraph 2, line 314-322)
Comment 8:
L289-290: The authors wrote "while 6 horses with a negative PCR result ... (either fever and/or ataxia). Does it mean that 6 out of 201 negative PCR showed clinical signs?Clarify the description.
Thanks for your comment, we hope it was clarified with the previous response (page 11, paragraph 2, line 314-322)
Comment 9:
L288-290: Did the horses show clinical signs before or after sample collection?
Thanks for your comment, we hope it was clarified with the previous response (page 11, paragraph 2, line 314-322)
Comment 10:
L298: The authors wrote "PCR analysis was ...". Which DNA were used for this PCR, blood or nasopharyngeal swabs? Clarify the description.
Thanks for your question. Sample type was now clarified: “Five horses showed titers of 1:128, three 1:64 and two >1:256. Subsequently, out of these cases, PCR analysis from nasopharyngeal swabs was positive in 3/13 cases (with titers of 1:32, 1:32, 1:64, respectively).” (page 11, paragraph 4, line 330-331).
Comment 11:
L300-335: The authors analyzed relationships between variables. How about fever and neurological signs. Add the results of relationships between fiver and neurological signs.
Thanks for your suggestion. The relationship between fever and neurological signs was now added: “Fever and neurological signs: A significant relationship was detected between the presence of fever and the development of neurological signs (EHM) (8/567 horses developed fever, 6/8 horses with fever developed EHM; chi square 179.895; p<0.001, Cramer's V 0.563)”. (Page 12, paragraph 3, line 360-363)
Comment 12:
L371: abortions8 is abortions [8].
Thanks for pointing this out, it was changed (page 13, paragraph 2, line 406)
Comment 13:
L394: (>80%)11 is (>80%) [11]. Thanks for noticing this mistake, it was corrected (page 13, paragraph 5, line 429)
Comment 14:
L413-414: Indicate individuals in Table 2 are 4 horses which developed EHM and never showed fever.
We appreciate your suggestion; it was added in Table 2. Those 4 horses are horses 3, 4, 5 and 10 (page 10)
Comment 15:
L418-420: the authors wrote "Absence of fever ... and serocovnerted". Are these descriptions about the present data or general things? If these are not about the present data, cite reference(s). If these are about the present data, indicate which individuals in Table 2.
We agree, it was not clarified. The description is about general EHV-1 data previously reported/studied. The reference to the description was now added: Absence of fever during EHV-1 infection correlates with no or low-grade viremia [15]. However, as primary respiratory tract infections occur, horses can be asymptomatic, but replicate and shed virus, and seroconvert [15]. (page 14, paragraph 1, line 453-455)
Comment 16:
L429: (5%)1 is (5%) [1]. Thanks for noticing this mistake, it was corrected (page 14, paragraph 2, line 465-466)
Comment 17:
L429: The authors wrote "... related to virus type". What does "virus type"mean? EHV-1 or EHV-4 or other things? Clarify the description.
Thanks for your comment, we hope it was now clarified: “This difference could be related to EHV-1 viral characteristics and prevalence or other risk factors”. (page 14, paragraph 2, line 464-4695)
Comment 18:
L433: "neuroprotection) The" is "neuroprotection. The". Thanks for noticing this mistake, it was corrected (page 14, paragraph 3, line 468)
Comment 19:
L445-446: Why was not EHV-1 ORF30 A/G/C2254 detection available? The authors should examine the PCR and sequencing to detect ORF30 A/G/C2254.
We agree. Sequencing has been attempted but has not been possible yet. Our central official lab team (SAG) is working hard to get it done, but there is very little genetic material in the collected samples. Probably because viral load in this outbreak was low (low CTs), and we have to add the fact that unfortunately the index case was not analyzed (the mare was euthanized and buried, it was thought it was an isolated case, probably due to trauma, once more cases started to appear, infectious diseases were suspected and investigations started). Orf30 PCR was requested by us, but primers were not available and purchased, unfortunately they have not arrived yet (the official lab from SAG is a governmental institution and bureaucracy can make processes sometimes a quite slow).

Round 2
Reviewer 2 Report
Comments and Suggestions for Authors
All of your replies to the comments are acceptable, including the comments about DNA sequences of ORF30 of the samples.